# Association of a healthy beverage score with total mortality in the adult population of Spain: A nationwide cohort study

Montserrat Rodríguez-Ayala[1,2], Carolina Donat-Vargas[1,3,4], Belén Moreno-Franco[5,6], Diana María Mérida[1], José Ramón Banegas[1], Fernando Rodríguez-Artalejo[1,7], Pilar Guallar-Castillón[1,7] *

1 Department of Preventive Medicine and Public Health, School of Medicine, Universidad Autónoma de Madrid and CIBERESP (CIBER of Epidemiology and Public Health), Madrid, Spain, 2 Department of Microbiology and Parasitology, Hospital Universitario La Paz, Madrid, Spain, 3 ISGlobal, Campus Mar., Barcelona, Spain, 4 Unit of Cardiovascular and Nutritional Epidemiology, Institute of Environmental Medicine, Karolinska Institutet, Stockholm, Sweden, 5 Instituto de Investigación Sanitaria (IIS) Aragón, Hospital Universitario Miguel Servet, Zaragoza, Spain, 6 CIBERCV (CIBER of Cardiovascular), Instituto de Salud Carlos III, Madrid, Spain, 7 IMDEA-Food Institute, CEI UAM+CSIC., Madrid, Spain

* mpilar.guallar@uam.es

**Data Availability Statement:** The data are freely available upon request by contacting Esther López-García at the Department of Preventive Medicine and Public Health, Faculty of Medicine, Universidad Autónoma de Madrid (UAM)-IdiPaz, CIBERESP (CIBER of Epidemiology and Public Health), 28029,

## Abstract

### Background

Despite the substantial evidence of the relationship between diet and mortality, the role of beverage consumption patterns is not well known. The aim of this study was to assess the association of the adherence to a Healthy Beverage Score (HBS) and all-cause mortality in a representative sample of the Spanish adult population.

### Methods and findings

We conducted an observational cohort study using data from the Study on Nutrition and Cardiovascular Risk in Spain (ENRICA), which included 12,161 community-dwelling individuals aged ≥18 years recruited in 2008 to 2010 and followed until January 2022. At baseline, food consumption was collected using a validated diet history. The HBS consists of 7 items, each of which is scored from 1 to 4 (highest adherence). The HBS ranges from 7 to 28 points with a higher score representing a healthier pattern. Adherence was assigned as a higher consumption of low-fat milk, and coffee and tea, a lower consumption of whole-fat milk, no consumption of fruit juice, artificially sweetened beverages, or sugar-sweetened beverages, and no or moderate consumption of alcohol. Total mortality was ascertained by linkage to the Spanish National Death Index. Statistical analyses were performed with Cox models and adjusted for the main confounders, including sociodemographic, lifestyle, dietary variables, and morbidity.

After a mean follow-up of 12.5 years (SD: 1.7; range: 0.5 to 12.9), a total of 967 deaths occurred. For all-cause mortality, the fully adjusted hazard ratio (HR) for the highest versus lowest sex-specific quartiles of HBS was 0.72 (95% confidence interval [0.57, 0.91], p linear-trend = 0.015), corresponding to an 8.3% reduction in the absolute risk of death. A linear

Madrid, Spain. E-mail address: esther.lopez@uam.
es. UAM website: https://www.uam.es/ss/Satellite/
Medicina/es/1242658444664/subhome/
Departamento_de_Medicina_Preventiva_y_Salud_
Publica_y_Microbiologia.htm.

**Funding:** This work was supported by FIS grants
17/1709, and 20/144 from the Carlos III Health
Institute, the Secretary of R+D+I, and the European
Regional Development Fund/European Social Fund
(to P.G-C); by the National Plan on Drugs grant
2020/17, Spanish Ministry of Health, Spain (to F.R-
A); by the FACINGLCOVID-CM project, Comunidad
de Madrid and European Regional Development
Fund (ERDF), European Union (to F.R-A); and by
the REACT EU Program, Comunidad de Madrid and
European Regional Development Fund (ERDF),
European Union (to F.R-A). The funders had no
role in study design, data collection and analysis,
decision to publish, or preparation of the
manuscript.

**Competing interests:** The authors have declared
that no competing interests exist.

**Abbreviations:** BMI, body mass index; CI,
confidence interval; ENRICA, Study on Nutrition
and Cardiovascular Risk in Spain; HBS, Healthy
Beverage Score; HR, hazard ratio; METs-hour/
week, metabolic equivalents in hours per week; SD,
standard deviation.

relationship between the risk of death and the adherence to the HBS was observed using
restricted cubic splines. The results were robust to sensitivity analyses. The main limitation
was that repeated measurements on beverage consumption were not available and bever-
age consumption could have changed during follow-up.

## Conclusions

In this study, we observed that higher adherence to the HBS was associated with lower total
mortality. Adherence to a healthy beverage pattern could play a role in the prevention of pre-
mature mortality.

## Author summary

### Why was this study done?

- Most dietary patterns focus solely on solid foods, and the role of beverages as a whole
  has received little attention.

- Our aim was to assess the relationship between a Healthy Beverage Score (HBS) and
  mortality in a representative sample of community-dwelling individuals from Spain.

- Our hypothesis was that high adherence to the HBS would be associated with lower
  mortality.

### What did the researchers do and find?

- We included a representative sample of 12,161 adults (18 years and older) from Spain
  who were recruited in 2008 to 2010 and followed up until 2022. A total of 967 deaths
  occurred.

- Participants were categorized according to their adherence to the HBS.

- A higher total score was achieved with a higher consumption of low-fat milk, and coffee
  and tea, no consumption of whole-fat milk, fruit juice, artificially sweetened beverages,
  sugar-sweetened beverages, and no consumption or moderate consumption of alcohol.

- Each HBS item scored from 1 (minimum adherence) to 4 points (maximum adherence)
  and the HBS ranged from 7 to 28 points. The higher the HBS, the healthier.

- When comparing extreme categories, higher adherence to the HBS was associated with
  lower all-cause mortality in the Spanish adult population, with an 8.3% reduction in the
  absolute risk of death.

### What do these findings mean?

- The adherence to the HBS could serve as a potential diet-based strategy to prevent pre-
  mature mortality.

- The quality of beverage patterns could influence health outcomes in the general population.

## Introduction

The influence of unhealthy dietary factors on adverse outcomes, including premature mortality, is a public health concern [1]. Thus, the association between diet and mortality has been examined by using several approaches such as analyzing individual nutrients, food and, more recently, assessing dietary patterns and indexes [2–4]. Most of the indexes include mainly solid food (e.g., meat, poultry, fish, as well as fruit and vegetables) [5], although some of them also comprise beverages [6].

The mechanisms by which beverages influence health are complex and are not only based on the nutritional quality of their components (such as energy provided, macronutrients, fiber, minerals, and vitamins) [7], but also rely on other factors such as satiety mechanisms and factors that affect the assimilation of beverages such as rapid gastric emptying and intestinal absorption [8]. Moreover, the addition of artificial sweeteners influences mortality [9]. On the other hand, some beverages are also an important source of other additives (e.g., phosphates) as well as contaminants from packaging or processing (e.g., organophosphate esters, phthalates) [10].

In 2015, a healthy beverage index, based on commonly consumed drinks, was developed to evaluate the role of beverage quality on cardiometabolic risk in adult Americans. A low adherence to this index was associated with several detrimental cardiometabolic markers [11]. Consistent results were obtained in another US study where the association between the adherence to a healthy beverage pattern and total mortality was evaluated. Data were obtained from a cohort of 2,283 adults, aged $\geq 21$ years, with a previous diagnosis of mild to moderate chronic renal insufficiency. A healthier beverage index was inversely associated with the progression of chronic kidney disease and all-cause mortality [12].

Although the association of specific beverages has been studied previously, the role of a healthy beverage index and its association with mortality has not been assessed in the general population yet. We hypothesized that higher adherence to a 7-item Healthy Beverage Score (HBS), previously proposed by Hu and colleagues [12] and adapted to the Spanish beverage consumption, could be associated with lower mortality. Therefore, the aim of this study is to assess the association between the HBS and all-cause mortality in a representative cohort of Spanish adults.

## Methods

### Study design and participants

Data were obtained from the Study on Nutrition and Cardiovascular Risk in Spain (ENRICA) whose methods have been reported elsewhere [13]. In brief, 13,105 individuals aged $\geq 18$ years were recruited from 2008 to 2010. A stratified cluster sampling based on the census sections of Spain was performed to guarantee the representativeness of the sample. Sample weights were based on the size of municipalities, sex, and age. Three sequential stages were followed for data collection. First, sociodemographic, lifestyle characteristics, and morbidity information was obtained through a telephone interview. Second, blood and urine samples were collected on a first home visit. Third, a physical examination and a face-to-face dietary history (DH-ENRICA) were completed during a second home visit. The response rate was 51% and the main reasons for non-participation were refusal to provide a blood sample (51.7%) and not being interested in the study (37.8%).

From the initial sample (13,105 individuals), 944 participants were excluded: 60 (0.5%) without information on diet and 884 (6.8%) with implausible values for total energy intake (<800 kcal/day or >5,000 kcal/day in males; <500 kcal/day or >4,000 kcal/day in females). Therefore, a total of 12,161 participants were included in the analysis (S1 Fig).

The Clinical Research Ethics Committee of La Paz University Hospital in Madrid provided ethical approval. All participants from the ENRICA Study gave written informed consent after explaining the details of the study.

## Study variables

**Dietary history.** Information on diet was obtained through a computerized dietary history (DH-ENRICA), conducted by trained and certified nonmedical interviewers. The DH-ENRICA collected information on 861 items of food, with 82 beverages included. Participants informed about all items of food and beverages consumed at least once every 2 weeks in the previous year. Food consumed during weekdays and weekends were considered. A total of 127 sets of digitalized photos, household measurements, as well as the usual proportion sizes of food from typical Spanish recipes were used to estimate portion sizes in grams per day. Regarding beverages, a total of 14 digitized photos and 23 household measurements were used to later estimate beverage consumption in milliliters per day. In addition, for alcoholic beverages, the consumption of ethanol in grams per day was calculated using Spanish food composition tables [14]. The validity correlation coefficients in HD-ENRICA for beverages were: 0.71 for coffee, 0.69 for milk, 0.40 for soft drinks, and 0.64 for alcoholic beverages [15].

**The Healthy Beverage Score (HBS).** A Healthy Beverage Score (HBS) was previously described by Hu and colleagues [12]. Based on the HBS, we built a 7-item HBS modifying its cut-off points to fit with the beverage consumption of a representative sample of the Spanish adult population. Each item of the HBS scored from 1 (minimal adherence) to 4 points (maximal adherence) based on sex-specific categories of consumption. Thus, the HBS ranged from 7 (low adherence) to 28 points (high adherence). The higher the HBS, the healthier the pattern. Items were grouped in 2 main components: adequacy and moderation (Table 1). Two beverages were considered as adequacy components: low-fat milk as well as coffee and tea consumption. For these 2 components, the higher the score, the healthier the pattern. No low-fat milk consumption scored 1, while the remaining sample was divided into tertiles among consumers; coffee and tea consumption was grouped into quartiles. Five items were included as moderation components: whole-fat milk, fruit juice, artificially sweetened beverages, sugar-sweetened beverages, and alcohol. For these 5 items, the higher the consumption, the lower the score, with a specific classification for alcohol consumption. Whole-fat milk and sugar-sweetened beverages scored 4 for no consumption and the remaining sample was divided into tertiles among consumers. Fruit juice and artificially sweetened beverages consumption scored 4 for no consumption and 1 for any consumption. The scoring of fruit juices and artificially sweetened beverages was decided on the basis of the lack of a wide range of consumption, and to maintain the relative weight of these items in the score, as previously described by Hu and colleagues [12]. Finally, for alcohol consumption, participants with no consumption or moderate drinkers (<40 g/day for males and <24 g/day for females) scored 4, and heavy drinkers (≥40 g/day for males and ≥24 g/day for females) scored 1.

**Mortality assessment.** All-cause mortality was ascertained through a computerized linkage with the Spanish National Death Index. Participants were followed from baseline in 2008 to 2010 to January 31, 2022. Follow-up was censored at the date of death or at the end of follow-up, whichever occurred first.

**Table 1. Scoring criteria for the HBS in the ENRICA Study (2008–2010).**

| Components | Minimum score | | | Maximum score |
|---|---|---|---|---|
| **Adequacy** | | | | |
| **Low-fat milk** | 1 (No consumption) | 2 (Tertile 1 among consumers) | 3 (Tertile 2 among consumers) | 4 (Tertile 3 among consumers) |
| **Coffee and tea** | 1 (Quartile 1) | 2 (Quartile 2) | 3 (Quartile 3) | 4 (Quartile 4) |
| **Moderation** | | | | |
| **Whole-fat milk** | 1 (Tertile 3 among consumers) | 2 (Tertile 2 among consumers) | 3 (Tertile 1 among consumers) | 4 (No consumption) |
| **Fruit juice** | 1 (Any consumption) | – | – | 4 (No consumption) |
| **Artificially sweetened beverages** | 1 (Any consumption) | – | – | 4 (No consumption) |
| **Sugar-sweetened beverages** | 1 (Tertile 3 among consumers) | 2 (Tertile 2 among consumers) | 3 (Tertile 1 among consumers) | 4 (No consumption) |
| **Alcohol**[a] | 1 (Heavy drinkers) | – | – | 4 (No consumption or moderate drinkers) |
| **Range** | **7** | | | **28** |

[a] Heavy drinkers were defined as consumption $\geq$40 g/day for males and $\geq$24 g/day for females. Among drinkers, a moderate alcohol consumption was defined as <40 g/day for males and <24 g/day for females.

**Confounders.** Participants provided information regarding age, sex, educational level, and smoking status which were obtained through the computer-assisted telephone interview performed at baseline. On the second home visit and following standardized procedures, blood pressure, and weight and height were measured. Body mass index (BMI) was calculated as weight divided by the square of the height in meters (kg/m$^2$). Leisure time and household physical activity were evaluated with the EPIC short questionnaire, collecting information on 17 activities. Then, each activity was multiplied by their respective energy expenditure rate in metabolic equivalents in hours per week (METs-hour/week) [16] and total energy expenditure was obtained by summing up all activities. Hours spent watching television were used to account for sedentary activities. In order to control for good dietary quality, analyses were adjusted for total energy intake, fiber, fruit and vegetable consumption, as well as the Mediterranean Diet Adherence defined by Trichopoulou and colleagues [17] without including alcohol. Blood samples collected on the first home visit were centrally analyzed in the CORE laboratory of La Paz University Hospital in Madrid. A colorimetric enzymatic method with lipase and glycerol kinase (for triglycerides) and a colorimetric enzymatic method with cholesterol-oxidase, esterase, and peroxidase (for cholesterol) were used. To define hypertriglyceridemia, we used a threshold of $\geq$150 mg/d in fasting plasma triglycerides levels, and for hypercholesterolemia, a fasting plasma total cholesterol level of $\geq$200 mg/dL or prescribed lipid-lowering medications. Hypertension was defined as $\geq$140/90 mmHg or taking antihypertensive medication. Analyses were also controlled for the number of chronic conditions (chronic obstructive pulmonary disease, coronary heart disease, stroke, heart failure, osteoarthritis, cancer, depression diagnosed by a physician, and diabetes), as well as the number of prescribed medications to consider prevalent morbidity.

Independent variables with missing values were imputed by using multiple imputation [18]: educational level (<1%), smoking status (<1%), BMI (1.5%), number of television hours (<1%), hypertriglyceridemia (<1%), hypercholesterolemia (<1%), and high blood pressure (1.1%). The validity of imputed data was examined against analyses performed with variables that contained full information.

## Statistical analysis

Across sex-specific quartiles of adherence to the HBS, age-adjusted baseline characteristics of participants were computed using marginals. Age-adjusted means for continuous variables and age-adjusted proportions (%) for categorical variables were provided. To estimate hazard ratios (HR) and their 95% confidence intervals (CIs), Cox proportional hazards regression models were built and age was considered as the underlying time metric. The survey command was applied to account for the complex sampling design. The lowest category of adherence to the HBS was used as reference.

Three sequential Cox models were used. Model 1 was an unadjusted model. Model 2 was additionally adjusted for age, sex, educational level, smoking status, ex-drinker status, BMI, physical activity in leisure time, total energy intake, fruit and vegetable consumption, total fiber intake, hypertriglyceridemia, hypercholesterolemia, hypertension, number of self-reported chronic conditions, and number of medications. Finally, Model 3 was adjusted for the Mediterranean index by Trichopoulou excluding alcohol (maximum score = 8), but excluding fruit, vegetable, and fiber consumption. Schoenfeld residuals were plotted against time to assess proportional hazards assumptions and visually no violations were found. To test for linear trend, categories of the HBS were modeled as a continuous variable. The dose–response relationship was assessed by using restricted cubic splines with 3 knots (at the 10th, 50th, and 90th percentiles). Interactions between the HBS and age (<65 years versus ≥65 years), sex (male versus female), BMI (<30 kg/m2 versus ≥30 kg/m2), physical activity (≤median 61.5 METs-hour/week versus >median 61.5 METs-hour/week), vegetable consumption (≤median 183.5 g/d versus >median 183.5 g/d), adherence to the Mediterranean diet without including alcohol (≤median 4 versus >median 4), and the prevalence of chronic conditions (yes/no) were tested by including multiplicative terms in Model 3. Sensitivity analyses were conducted excluding deaths in the first 3 years of follow-up to account for the effect of subclinical conditions at baseline. Also, individual items of the HBS were assessed according to Model 3 plus adjustment for the remaining items that are part of the score.

This was a preplanned study and data analysis was conducted according to a prespecified plan (S1 Text). The HBS was previously used to assess the association between adherence to HBS and age-related frailty in a sample of Spanish older adults [19]. A minor modification was introduced in alcohol consumption classification. For older adults, moderate alcohol consumption was considered the healthy option due to their high cardiovascular risk [19]. However, in the current analysis, which involves the adult general population (aged ≥18 years) with lower cardiovascular risk, the category of alcohol considered healthy was "no consumption or moderate consumption."

This study was reported as per the Strengthening the Reporting of Observational Studies in Epidemiology (STROBE) guideline (S1 STROBE Checklist).

A two-tailed *p* value less than 0.05 was considered statistically significant. All analyses were performed with Stata, version 17.0 (StataCorp, College Station, Texas, United States of America).

## Results

The median age of participants (*N* = 12,161) was 46 years old (interquartile range 35 to 61) and 52.6% were females. Compared with those in quartile 1 (less healthy) of adherence to the HBS, participants in quartile 4 (healthier) were older, more frequently females, with a higher level of education and with a less sedentary lifestyle, and were more physically active. Also, those in quartile 4 had lower energy intake, consumed more fiber, fruit and vegetables, showed

**Table 2. Age-adjusted baseline characteristics of participants in the ENRICA Study (2008–2010) by quartiles of the HBS (N = 12,161).**

| Characteristics | Quartiles of adherence to the HBS | | | | p for trend |
|---|---|---|---|---|---|
| | Quartile 1 (Less healthy) n = 2,813 | Quartile 2 n = 2,985 | Quartile 3 n = 2,745 | Quartile 4 (Healthier) n = 3,618 | |
| Age, mean, y | 38.9 | 45.3 | 48.5 | 53.9 | <0.001 |
| Female, % | 53.4 | 39.3 | 50.7 | 57.9 | <0.001 |
| Educational level, % | | | | | <0.001 |
| Primary or less | 29.5 | 24.3 | 23.8 | 27.2 | |
| Secondary | 43.8 | 46.1 | 43.2 | 40.0 | |
| University | 26.7 | 29.6 | 32.9 | 32.8 | |
| Smoking, % | | | | | <0.001 |
| Non-smoker | 53.4 | 45.6 | 47.3 | 49.6 | |
| Former smoker | 20.1 | 25.8 | 25.6 | 26.5 | |
| Current smoker | 26.5 | 28.6 | 27.1 | 23.9 | |
| Ex-drinker, % | 56.1 | 50.0 | 52.1 | 48.3 | 0.320 |
| BMI, % | | | | | <0.001 |
| $<25$ kg/m$^2$ | 39.9 | 33.2 | 35.6 | 38.0 | |
| 25-$<30$ kg/m$^2$ | 38.6 | 43.3 | 42.9 | 39.5 | |
| $\geq 30$ kg/m$^2$ | 21.6 | 23.6 | 21.5 | 22.5 | |
| Time watching TV, mean, h | 2.1 | 1.9 | 1.9 | 1.9 | 0.005 |
| Physical activity, mean, METs-hour/week | 67.0 | 65.4 | 68.3 | 72.1 | <0.001 |
| Energy consumption, mean, Kcal/d | 2,331.9 | 2,261.9 | 2,135.5 | 1,992.8 | <0.001 |
| Fiber consumption, mean, g/d | 22.5 | 22.9 | 23.2 | 23.0 | 0.035 |
| Fruit consumption, mean, g/d | 222.8 | 233.7 | 245.5 | 247.5 | <0.001 |
| Vegetable consumption, mean, g/d | 184.0 | 197.2 | 211.6 | 209.6 | <0.001 |
| Mediterranean diet score (without alcohol), mean | 3.7 | 3.9 | 4.1 | 4.0 | <0.001 |
| Hypertriglyceridemia, % | 17.8 | 19.8 | 18.3 | 16.8 | 0.127 |
| Hypercholesterolemia, % | 46.9 | 50.5 | 52.7 | 52.8 | <0.001 |
| Hypertension, % | 29.2 | 32.2 | 27.6 | 28.3 | 0.109 |
| Number of chronic conditions[a], % | | | | | <0.001 |
| None | 71.9 | 74.8 | 71.7 | 69.0 | <0.001 |
| One | 22.3 | 20.5 | 22.2 | 24.7 | |
| Two or more | 5.8 | 4.6 | 6.1 | 6.3 | |
| Number of medications, % | | | | | <0.001 |
| None | 72.0 | 69.4 | 71.4 | 71.0 | |
| One to 3 | 25.4 | 27.6 | 24.5 | 25.5 | |
| More than 3 | 2.6 | 3.0 | 4.1 | 3.5 | |

[a] Chronic conditions included: chronic obstructive pulmonary disease, coronary heart disease, stroke, heart failure, osteoarthritis, cancer, depression diagnosed by a physician, and diabetes.

HBS, Healthy Beverage Score; BMI, body mass index; METs-hour/week, metabolic equivalents in hours per week.

a higher adherence to the Mediterranean diet, and had more frequently hypercholesterolemia (Table 2). Sex-specific cut-off points for individual items of the HBS are shown in S1 Table.

In accordance with the rules for the construction of the HBS, compared with those in quartile 1 (less healthy), participants in quartile 4 (healthier) consumed more low-fat milk, coffee and tea, and alcohol, but consumed less whole-fat milk, fruit juice, artificially sweetened beverages, and sugar-sweetened beverages (Table 3).

**Table 3. Beverage consumption by quartiles of the HBS in the ENRICA Study (2008–2010) (N = 12,161).**

| HBS components | Quartiles of adherence to the HBS[a] | | | | p value |
|---|---|---|---|---|---|
| | Quartile 1 (Less healthy) n = 2,813 | Quartile 2 n = 2,985 | Quartile 3 n = 2,745 | Quartile 4 (Healthier) n = 3,618 | |
| **Adequacy** | | | | | |
| Low-fat milk, mean (SD), mL/d | 50.0 (92.7) | 100.3 (128.2) | 149.3 (155.1) | 215.5 (148.3) | <0.001 |
| Coffee and tea, mean (SD), mL/d | 64.3 (88.2) | 103.1 (121.8) | 127.5 (138.8) | 161.5 (154.6) | <0.001 |
| **Moderation** | | | | | |
| Whole-fat milk, mean (SD), mL/d | 139.3 (159.3) | 82.8 (121.6) | 38.2 (76.4) | 8.8 (23.7) | <0.001 |
| Fruit juice, mean (SD), mL/d | 100.4 (139.9) | 66.1 (124.2) | 36.9 (87.7) | 7.6 (42.6) | <0.001 |
| Artificially sweetened beverages, mean (SD), mL/d | 48.9 (156.2) | 34.5 (219.7) | 17.7 (115.0) | 3.5 (51.3) | <0.001 |
| Sugar-sweetened beverages, mean (SD), mL/d | 162.0 (249.4) | 71.0 (168.2) | 43.2 (125.7) | 10.2 (68.7) | <0.001 |
| Alcohol, mean (SD), g/d[b] | 11.4 (19.2) | 10.7 (17.8) | 7.9 (14.0) | 5.8 (10.4) | <0.001 |

[a] Cut-off points for the HBS = for males: Q1 10–18; Q2 19–21; Q3 22–23; Q4 24–28; for females: Q1 10–19; Q2 20–21; Q3 22–23; Q4 24–28.

[b] Alcohol was defined as the consumption of ethanol in grams per day.

HBS, Healthy Beverage Score; SD, standard deviation.

After a mean follow-up of 12.5 years (SD: 1.7; range: 0.5 to 12.9) and 151,459 person-years of follow-up, a total of 967 deaths occurred. The HR for all-cause mortality when comparing extreme quartiles of the adherence to the HBS was 0.72 (95% CI, 0.57 to 0.91, p for linear trend = 0.015) in the fully adjusted model (Table 4). The decrease in absolute risk of death was 4.3% for quartile 2, 6.3% for quartile 3, and 8.3% for quartile 4. No significant interactions were found for age, sex, BMI, physical activity, vegetable consumption, or adherence to the Mediterranean diet (without including alcohol). However, a statistically significant interaction was found when stratifying for the presence of at least 1 chronic condition (p = 0.030). Among those with at least 1 chronic condition, higher adherence to the HBS was associated with lower mortality. No association was observed among those with no chronic conditions (Table 5).

**Table 4. Mortality risk according to quartiles of the adherence to the HBS in the ENRICA Study from baseline (2008–2010) to January 2022 (N = 12,161).**

| Total mortality | Quartile 1 HR (95% CI) (Less healthy) | Quartile 2 HR (95% CI) | Quartile 3 HR (95% CI) | Quartile 4 HR (95% CI) (Healthier) | p for linear trend[d] |
|---|---|---|---|---|---|
| Deaths/n | 141/2,813 | 227/2,985 | 228/2,745 | 371/3,618 | |
| Person-years | 36,216 | 38,058 | 32,860 | 44,325 | |
| Model 1[a] | 1 (ref.) | 0.86 [0.68, 1.10] | 0.84 [0.65, 1.08] | 0.75 [0.59, 0.94] | 0.011 |
| Model 2[b] | 1 (ref.) | 0.79 [0.61, 1.02] | 0.77 [0.59, 1.00] | 0.72 [0.57, 0.92] | 0.017 |
| Model 3[c] | 1 (ref.) | 0.79 [0.61, 1.02] | 0.78 [0.60, 1.02] | 0.72 [0.57, 0.91] | 0.015 |

[a]Model 1 was an unadjusted model. Age was the underlying time metric.

[b]Model 2 was adjusted for age (years, continuous), sex (male, female), educational level (primary or less, secondary, university), smoking (non-smoker, former smoker, current smoker), ex-drinker (yes/no), BMI (<25, ≥25 and ≤30, >30 kg/m$^2$), time watching TV (hours, continuous), physical activity (METs-hour/week, continuous), energy intake (kcal/day, continuous), fiber intake (g/d continuous), fruit and vegetable consumption (g/d, continuous), hypertriglyceridemia (yes/no), hypercholesterolemia (yes/no), hypertension (yes/no), number of chronic conditions (0, 1, and ≥2), and number of medications (0, 1–3, >3). Age was the underlying time metric.

[c]Model 3 was adjusted for factors in Model 2 plus adherence to the Mediterranean diet without including alcohol (maximum score = 8) and excluding fruit, vegetable, and fiber consumption. Age was the underlying time metric.

[d]p value for quartile 4 vs. quartile 1: Model 1 p = 0.012, Model 2 p = 0.007; Model 3 p = 0.007.

BMI; body mass index; CI, confidence interval; HBS, Healthy Beverage Score; HR, hazard ratio.

**Table 5. Mortality risk according to quartiles of the adherence to the HBS in the ENRICA Study from baseline (2008–2010) to January 2022 by age, sex, BMI, physical activity, vegetable consumption, adherence to the Mediterranean diet without including alcohol and prevalence of chronic conditions (N = 12,161).**

| Total mortality | Quartile 1 HR (95% CI) (Less healthy) | Quartile 2 HR (95% CI) | Quartile 3 HR (95% CI) | Quartile 4 HR (95% CI) (Healthier) | p for linear trend | p for interaction[a] |
|---|---|---|---|---|---|---|
| **Age** | | | | | | |
| **<65 years, n = 9,774** | | | | | | |
| Deaths, n | 39/2,514 | 67/2,486 | 58/2,193 | 79/2,581 | | |
| Model 3[b] | 1 (ref.) | 1.19 [0.74, 1.90] | 1.11 [0.70, 1.78] | 0.99 [0.64, 1.54] | 0.680 | 0.364 |
| **≥65 years, n = 2,387** | | | | | | |
| Deaths, n | 102/299 | 160/499 | 170/552 | 292/1,037 | | |
| Model 3[b] | 1 (ref.) | 0.71 [0.53, 0.95] | 0.73 [0.53, 0.99] | 0.68 [0.51, 0.89] | 0.025 | |
| **Sex** | | | | | | |
| **Male, n = 5,760** | | | | | | |
| Deaths, n | 63/1,243 | 141/1,750 | 125/1,294 | 206/1,473 | | |
| Model 3[b] | 1 (ref.) | 0.93 [0.65, 1.33] | 0.90 [0.62, 1.31] | 0.89 [0.63, 1.24] | 0.482 | 0.287 |
| **Female, n = 6,401** | | | | | | |
| Deaths, n | 78/1,570 | 86/1,235 | 103/1,451 | 165/2,145 | | |
| Model 3[b] | 1 (ref.) | 0.67 [0.47, 0.97] | 0.68 [0.47, 0.98] | 0.58 [0.42, 0.81] | 0.004 | |
| **BMI** | | | | | | |
| **<30 kg/m², n = 9,513** | | | | | | |
| Deaths, n | 104/2,332 | 144/2,338 | 158/2,148 | 244/2,695 | | |
| Model 3[b] | 1 (ref.) | 0.80 [0.60, 1.07] | 0.82 [0.61, 1.10] | 0.72 [0.55, 0.95] | 0.038 | 0.932 |
| **≥30 kg/m², n = 2,648** | | | | | | |
| Deaths, n | 37/481 | 83/647 | 70/597 | 127/923 | | |
| Model 3[b] | 1 (ref.) | 0.72 [0.44, 1.18] | 0.65 [0.38, 1.09] | 0.65 [0.40, 1.05] | 0.144 | |
| **Physical activity** | | | | | | |
| **≤Median (61.5 METs-hour/week), n = 6,082** | | | | | | |
| Deaths, n | 89/1,451 | 142/1,557 | 148/1,373 | 246/1,701 | | |
| Model 3[b] | 1 (ref.) | 0.79 [0.58, 1.09] | 0.80 [0.59, 1.09] | 0.76 [0.58, 1.02] | 0.140 | 0.603 |
| **>Median (61.5 METs-hour/week), n = 6,079** | | | | | | |
| Deaths, n | 52/1,362 | 85/1,428 | 80/1,372 | 125/1,917 | | |
| Model 3[b] | 1 (ref.) | 0.84 [0.55, 1.27] | 0.79 [0.50, 1.26] | 0.65 [0.43, 0.99] | 0.038 | |
| **Vegetable consumption** | | | | | | |
| **≤Median (183.5 g/d), n = 6,083** | | | | | | |
| Deaths, n | 72/1,576 | 121/1,532 | 125/1,287 | 198/1,688 | | |
| Model 3[b] | 1 (ref.) | 82 [0.58, 1.17] | 0.93 [0.65, 1.32] | 0.75 [0.54, 1.03] | 0.113 | 0.284 |
| **>Median (183.5 g/d), n = 6,078** | | | | | | |
| Deaths, n | 69/1,237 | 106/1,453 | 103/1,458 | 173/1,930 | | |
| Model 3[b] | 1 (ref.) | 0.71 [0.50, 1.02] | 0.58 [0.39, 0.85] | 0.64 [0.45, 0.92] | 0.043 | |
| **Adherence to the Mediterranean diet without including alcohol** | | | | | | |
| **≤Median (4), n = 7,400** | | | | | | |
| Deaths, n | 82/2,002 | 123/1,843 | 117/1,557 | 181/1,998 | | |
| Model 3[b] | 1 (ref.) | 0.84 [0.59, 1.19] | 0.94 [0.66, 1.35] | 0.77 [0.55, 1.07] | 0.175 | 0.325 |
| **>Median (4), n = 4,761** | | | | | | |
| Deaths, n | 59/811 | 104/1,142 | 111/1,188 | 190/1,620 | | |
| Model 3[b] | 1 (ref.) | 0.74 [0.51, 1.08] | 0.63 [0.43, 0.90] | 0.66 [0.47, 0.92] | 0.033 | |

(*Continued*)

**Table 5.** (Continued)

| Total mortality | Quartile 1 HR (95% CI) (Less healthy) | Quartile 2 HR (95% CI) | Quartile 3 HR (95% CI) | Quartile 4 HR (95% CI) (Healthier) | p for linear trend | p for interaction[a] |
|---|---|---|---|---|---|---|
| **Prevalence of chronic conditions** | | | | | | |
| **No, n = 8,151** | | | | | | |
| Deaths, n | 45/2,143 | 82/2,124 | 70/1,811 | 114/2,073 | | |
| Model 3[b] | 1 (ref.) | 1.15 [0.73, 1.81] | 1.18 [0.75, 1.87] | 1.16 [0.74, 1.81] | 0.616 | 0.030 |
| **Yes, n = 4,010** | | | | | | |
| Deaths, n | 96/670 | 145/861 | 158/934 | 257/1,545 | | |
| Model 3[b] | 1 (ref.) | 0.67 [0.49, 0.91] | 0.63 [0.46, 0.86] | 0.57 [0.43, 0.76] | <0.001 | |

[a]p for interaction was calculated using the Wald test.

[b]Model 3 was adjusted for age (years, continuous), sex (male, female), educational level (primary or less, secondary, university), smoking (non-smoker, former smoker, current smoker), ex-drinker (yes/no), BMI (<25, $\geq$25 and $\leq$30, >30 kg/m$^2$), time watching TV (hours, continuous), physical activity (METs-hour/week, continuous), energy intake (kcal/day, continuous), hypertriglyceridemia (yes/no), hypercholesterolemia (yes/no), hypertension (yes/no), number of chronic conditions (0, 1, and $\geq$2), number of medications (0, 1–3, >3), adherence to the Mediterranean diet without including alcohol (maximum score = 8) as appropriate. Age was the underlying time metric.

BMI, body mass index; CI, confidence interval; HBS, Healthy Beverage Score; HR, hazard ratio; METs-hour/week, metabolic equivalents in hours per week.

After excluding the first 3 years of follow-up, the inverse association between the adherence to the HBS and total mortality remained similar (S2 Table). When assessing dose–response, a linear relationship was observed using restricted cubic splines (p value for non-linearity = 0.010) (Fig 1).

When individual HBS items were analyzed using Model 3 for adjustment, a higher consumption of coffee and tea, and no consumption of fruit juices and artificially sweetened beverages contributed most to the association with lower all-cause mortality (Fig 2). Unadjusted results are also shown (S3 Fig).

## Discussion

In this large population-based study of Spanish adults with a mean follow-up of 12.5 years, a higher adherence to the HBS was inversely associated with total mortality, after adjusting for potential confounders. Those with higher adherence to the HBS had an 8.3% reduction in the absolute risk of death compared to those with lower adherence. The association was linear and robust. It may also be of particular interest to people with preexisting chronic conditions, as they had lower mortality, although these findings need to be confirmed in future research.

Regarding to items of the HBS, 2 recent prospective studies performed in Spain found an inverse association between coffee consumption and all-cause mortality [20,21]. Results from the EPIC study (with 500,000 participants from 10 European countries) [22] and from the UK Biobank study were also similar to the findings in this study [23]. Our results are also in line with meta-analyses comprising cross-sectional studies, longitudinal cohorts, as well as interventional studies [24–26]. The beneficial effect of coffee might rely, among others, on the antioxidant and anti-inflammatory activity exhibited by its bioactive components, mainly melanoidins, chlorogenic acids, and caffeine [27]. These compounds reduce oxidative stress and inflammation [28], enhance endothelial function [29], and counteract carcinogenesis on in vitro studies [30]. Coffee also increases the metabolic rate [31], improves the glucose metabolism [32], and lowers long-term blood pressure [33]. Moreover, coffee could reduce mortality even in those with impaired caffeine metabolism [34] and independently to the addition of sweeteners [35]. However, high coffee consumption has been associated with an increase in

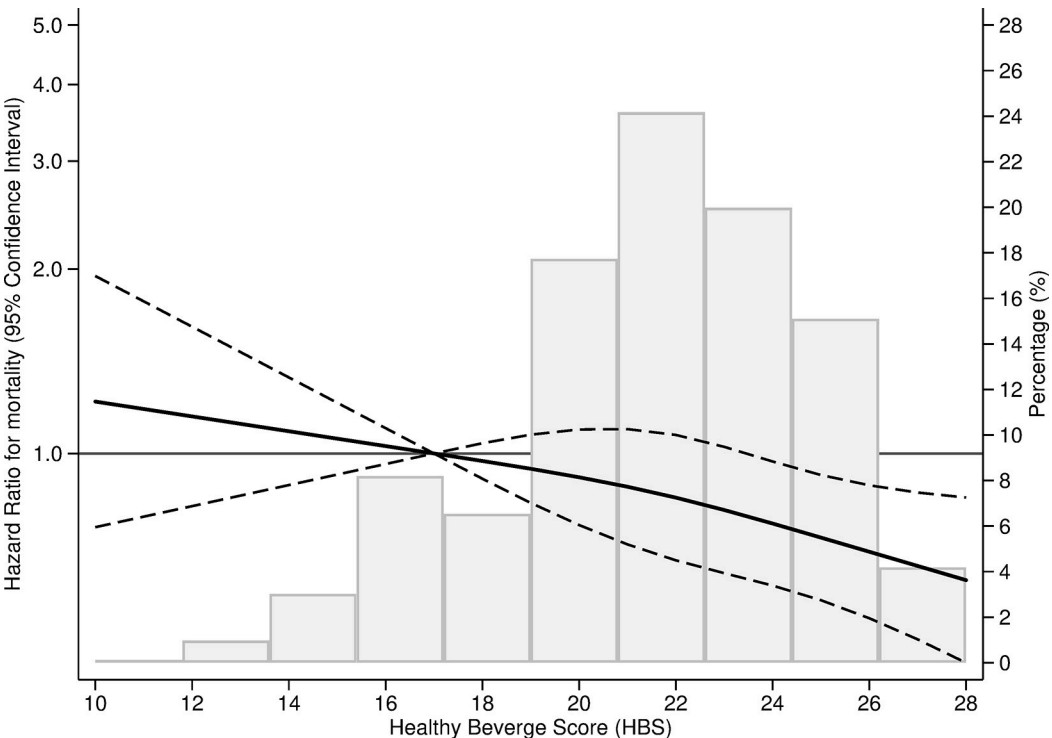

**Fig 1. Adjusted restricted cubic splines of the association of the HBS with mortality risk in the ENRICA Study from baseline (2008–2010) to January 2022 ($N$ = 12,161).** Lines are restricted cubic splines, showing the dose–response association of the HBS with mortality. The solid line represents the HR, and the dashed lines indicate the lower and upper 95% CIs. The knots were located at the 10th, 50th, and 90th percentiles (corresponding to HBS scores 17, 22, and 25, respectively); $p$ for non-linearity = 0.010. Adjusted as in Model 3. Data were adjusted for age (years, continuous), sex (male, female), educational level (primary or less, secondary, university), smoking (non-smoker, former smoker, current smoker), ex-drinker (yes/no), BMI ($<25$, $\geq25$ and $\leq30$, $>30$ kg/m$^2$), time watching TV (hours, continuous), physical activity (METs-hour/week, continuous), energy intake (kcal/day, continuous), hypertriglyceridemia (yes/no), hypercholesterolemia (yes/no), hypertension (yes/no), number of chronic conditions (0, 1, and $\geq2$), number of medications (0, 1–3, >3), and adherence to the Mediterranean diet without including alcohol (maximum score = 8). Age was the underlying time metric.

serum levels of total cholesterol, LDL-cholesterol, and triglycerides [36]. On the other hand, evidence suggests that coffee consumption above 4 cups/day is not associated with further lower mortality [25].

Spain is included among the European countries with the lowest tea consumption [37] and we did not find studies that evaluated its relationship with mortality among Spanish adults. Therefore, it is unlikely that tea consumption accounts for our results. However, in literature, both all-cause and cardiovascular mortality were reduced among tea consumers [38].

The effect of milk on health has been widely studied due to its fatty acid composition [39]. Two recent cohort studies showed that low-fat milk consumption was associated with lower all-cause mortality when compared to whole-fat milk consumption [40,41]. Whole-fat milk has a higher content of saturated fats that has been related to an increase in LDL-cholesterol and atherosclerosis [42]. As a result, whole-fat milk consumption could be particularly harmful among individuals with known cardiovascular risk. However, in a clinical trial among normo-cholesterolemic individuals, whole-fat milk consumption showed no impairment in lipid profile nor in glucose-insulin metabolism when compared to low-fat milk consumption [43]. It is of note that, milk could modulate satiety mechanisms [44] and also has several components with potential beneficial effects, such as caseins with antioxidant properties [45]. Milk is also rich in minerals, vitamins, and other bioactive compounds involved in anti-inflammatory and

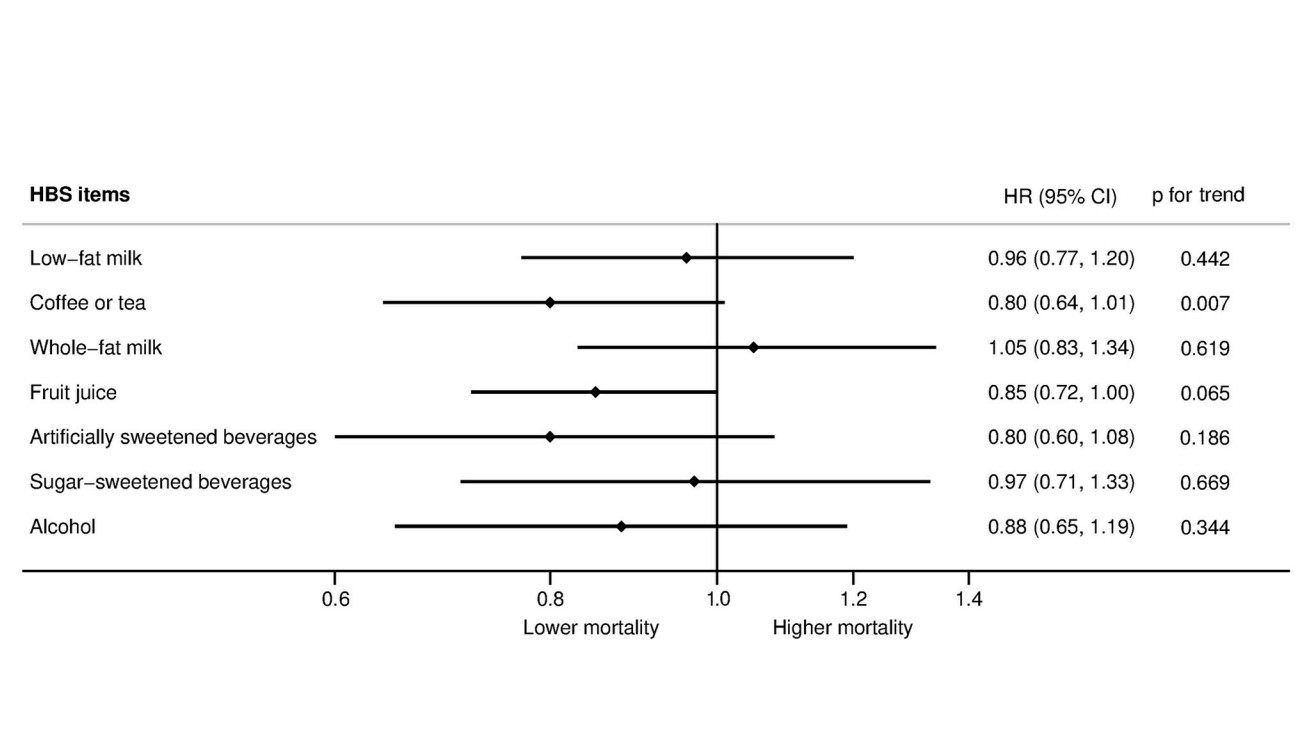

**Fig 2. Adjusted mortality risk for individual items of the HBS when comparing extreme categories (quartile 4 vs. quartile 1) in the ENRICA Study from baseline (2008–2010) to January 2022 ($N$ = 12,161).** Adjusted as in Model 3. Data were adjusted for age (years, continuous), sex (male, female), educational level (primary or less, secondary, university), smoking (non-smoker, former smoker, current smoker), ex-drinker (yes/no), BMI ($<25$, $\geq25$ and $\leq30$, $>30$ kg/m$^2$), time watching TV (hours, continuous), physical activity (METs-hour/week, continuous), energy intake (kcal/day, continuous), hypertriglyceridemia (yes/no), hypercholesterolemia (yes/no), hypertension (yes/no), number of chronic conditions (0, 1, and $\geq2$), number of medications (0, 1–3, $>3$), adherence to the Mediterranean diet without including alcohol (maximum score = 8), and for the rest of items of the HBS (as appropriate). Age was the underlying time metric. CI, confidence interval; HBS, Healthy Beverage Score; HR, hazard ratio.

immune regulation [46]. A recent meta-analysis showed that whole-fat milk consumption was associated with a higher risk of all-cause, cardiovascular disease and cancer mortality; however, low-fat milk showed a protective but nonsignificant association [47].

Results on fruit juice consumption and mortality mostly depend on the distinction between processed or fresh fruit juice [48]. A recent meta-analysis showed that processed fruit juice consumption was associated with a higher risk of type 2 diabetes and total mortality [49]. Evidence on fresh fruit juice consumption and health, however, is insufficient to draw conclusions. Results of a cohort study from the US with 13,440 participants showed that a higher consumption of 100% fruit juice was associated with a higher mortality [50]. Conversely, a study with 198,285 individuals from the UK found a positive association between sugar-sweetened beverages and mortality, but not for 100% fruit juice consumption [51]. Similarly, a recent meta-analysis of prospective cohorts concluded that there was no association between 100% fruit juice consumption and all-cause mortality [52]. Several studies have also found

that, compared with 100% fruit juice, sugary or processed fruit beverages produce harmful glucose levels after ingestion, mainly due to the higher content of free sugars [53]. In our study, bottled, sweetened, as well as fresh fruit juices were analyzed together as a unique item because of their rapid absorption [8] and similar effect on postprandial glucose levels [54]. Fructose intake, particularly from sugar-sweetened beverages at any dose, or from fruit juice at higher doses, contributes a rapid extra dietary energy source that could explain its detrimental effect on health [55]. However, food-based dietary guidelines from various countries from Europe Union, including Spain, consent to replace occasionally 1 daily portion of fruit with fresh fruit juice [56,57].

In order to lower calorie intake and control body weight, artificially sweetened beverages could be adequate short-term substitutes [58]. However, when considering the long-term influence of artificially sweetened beverage consumption, several studies have found associations with higher obesity, hypertension, type 2 diabetes, stroke, cardiovascular disease incidence and mortality, and all-cause mortality [59–61]. Since these beverages contain few to no calories nor sugars [51], some investigations have related them with weight gain as a result of an increased consumption of sweet food due to a greater affinity for sweet flavors or the perception of eating fewer calories [62]. In addition, their flavoring components have been associated with the formation of advanced glycation end-products [63], which are involved in the development of metabolic diseases [64]. Moreover, some sweeteners such as sucralose and saccharin could induce glucose intolerance and alterations in gut microbiota [65] that are linked to obesity [66].

In literature, a low to moderate alcohol consumption is related to a reduction in all-cause mortality [67]. Biological explanations for this protective role on health are based on lipid regulation, insulin response, and endothelial function [68] resulting from the modulation of some anti-inflammatory biomarkers [69]. However, at high doses, alcohol is detrimental to cardiovascular health and is related to several types of cancer [70]. A harmful alcohol consumption is associated to neurodegenerative processes [71], microbial dysbiosis [70], and an increased intestinal permeability that leads to a permanent hepatic exposure to bacterial translocation, oxidative stress, and other inflammatory components [72]. In addition, alcohol use could result in hepatic steatosis and de novo lipogenesis, and also could reduce the utilization of lipids [73]. Lastly, alcohol may injure myocardium with potential cardiomyopathy and heart failure [74], and increase the risk of hypertension [75]. On the other hand, the beneficial association of alcohol consumption with mortality found in some studies may rely on abstinence bias, insufficient adjustment for covariates, or consumption changes due to disease detection [76,77]. However, recent studies from an epigenetic perspective have proposed that alcohol at restricted doses could be particularly beneficial in older adults based on changes in alcohol metabolism related to age [78]. In this regard, a meta-analysis for the Global Burden of Disease Study has proposed a change from sex-specific to age-specific recommendations on alcohol consumption [79]. Then, low alcohol consumption could be beneficial among older adults, but not for younger adults. For our analyses, we considered that being a heavy drinker was harmful because of its well-established association.

In a previous study, a 10-item Healthy Beverage Index was constructed using an a priori approach based on the US recommendations for beverage consumption [11]. Similar to this index, Hu and colleagues described the HBS as a more suitable score for use in large epidemiological studies [12]. The HBS excluded water consumption as well as 2 items on total energy from beverages and calculations of daily fluid intake. In our study, the same scoring weights (from 1 to 4) were maintained for all items in the HBS. However, in contrast to Hu and colleagues, we considered both no alcohol consumption and moderate alcohol consumption as healthy. Additionally, we also modified the HBS cut-off points of the items to fit with the

beverage consumption of the Spanish adult population. As a result, the HBS used in our study retained the same items as originally described by Hu and colleagues, as well as the relative weight of the items.

The use of the HBS as an overall measure of beverage consumption has several advantages. First, the use of this score overcomes the limitations of analyses of relationships between individual beverages and diseases, as beverage consumption may be correlated, and an increase in consumption of one beverage may be associated with a decrease in the others. Secondly, the HBS could be a complementary tool for assessing adherence to dietary patterns that include only solid foods, in order to assess dietary quality as a whole. Thus, the HBS could serve as a simple and rapid screener to obtain information on the quality of beverage consumption from the general population, similar to other indexes used to assess adherence to certain diets, such as the Mediterranean diet. Also, the use of this 7-item pattern may be an optimal choice when dealing with patients in time-constrained clinical settings. Finally, the HBS includes items on commonly consumed beverages and could be easily adapted to other populations with only minor modifications to account for their specific beverage consumption.

We have used the HBS in the general population and caution should be exercised in deriving beverage consumption recommendations from this score in specific populations, especially those with restricted fluid requirements, long-term liquid diets, and other preexisting conditions involving fluid consumption. Further studies are also needed in specific population subgroups. In addition, a future study could consider intercorrelations and specific population-based patterns of beverage consumption using an a posteriori approach.

This study has some limitations. First, when measuring diet, non-differential misclassification of the exposure is always possible, in general, resulting in an underestimation of the associations found. Second, no information concerning behavioral changes or repeated measurements on beverage consumption were available, and beverage consumption could have changed during follow-up. Third, water consumption was not available in this study. Water is universally recommended as a safe beverage and as the main source of hydration. As water does not provide energy, macronutrients or micronutrients, its consumption is considered free for the general population.

There are also some strengths. To our knowledge, this is the first examination on the relation between a healthy beverage score and all-cause mortality among the Spanish adult population. In addition, we used a dietary history that allowed us to collect information on beverages with validity and reproducibility in a Spanish population. Also, the national vital statistics records, accessed through linkage to the Spanish National Death Index, ensured an extensive follow-up of the cohort for mortality assessment. Finally, several confounders were considered in more adjusted models.

In conclusion, in this representative study of the Spanish adult population, higher adherence to the HBS was associated with a reduction in total mortality. As the consumption of a healthy solid diet should be encouraged, adherence to a healthy beverage consumption pattern may also play an important role in the prevention of premature mortality as part of public health nutrition prevention strategies.

## Supporting information

**S1 STROBE Checklist. STROBE Checklist.**
(DOCX)

**S1 Text. Prespecified analysis plan and modifications.**
(DOCX)

**S1 Table. Sex-specific cut-off points for individual items of the Healthy Beverage Score (HBS) in the ENRICA Study (2008–2010) (N = 12,161).**
(DOCX)

**S2 Table. Mortality risk according to quartiles of the adherence to the Healthy Beverage Score (HBS) in the ENRICA Study from baseline (2008–2010) to January 2022 (N = 12,161) excluding the first 3 years of follow-up.**
(DOCX)

**S1 Fig. Flow diagram.**
(TIF)

**S2 Fig. Unadjusted restricted cubic splines of the association of the Healthy Beverage Score (HBS) with mortality risk in the ENRICA Study from baseline (2008-2010) to January 2022 (N = 12,161).** Lines are restricted cubic splines, showing the dose-response association of the Healthy Beverage Score (HBS) with mortality. The solid line represents the hazard ratio (HR), and the dashed lines indicate the lower and upper 95% confidence intervals. The knots were located at the 10th, 50th, and 90th percentiles (corresponding to HBS scores 17, 22 and 25, respectively). p for non-linearity = 0.003.
(TIF)

**S3 Fig. Unadjusted mortality risk for individual items of the Healthy Beverage Score (HBS) when comparing extreme categories (quartile 4 vs. quartile 1) in the ENRICA Study from baseline (2008-2010) to January 2022 (N = 12,161).** HBS, Healthy Beverage Score; HR, hazard ratio; CI, confidence interval.
(TIF)

## Author Contributions

**Conceptualization:** Pilar Guallar-Castillón.

**Data curation:** Montserrat Rodríguez-Ayala.

**Formal analysis:** Montserrat Rodríguez-Ayala.

**Funding acquisition:** Fernando Rodríguez-Artalejo, Pilar Guallar-Castillón.

**Investigation:** Montserrat Rodríguez-Ayala.

**Methodology:** Montserrat Rodríguez-Ayala, Pilar Guallar-Castillón.

**Software:** Montserrat Rodríguez-Ayala.

**Supervision:** Pilar Guallar-Castillón.

**Validation:** Pilar Guallar-Castillón.

**Visualization:** Montserrat Rodríguez-Ayala.

**Writing – original draft:** Montserrat Rodríguez-Ayala.

**Writing – review & editing:** Carolina Donat-Vargas, Belén Moreno-Franco, Diana María Mérida, José Ramón Banegas, Fernando Rodríguez-Artalejo.

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
