## [Editor Report · Decision Letter 0]

14 Jul 2023

Dear Dr Guallar-Castillón, 

Thank you for submitting your manuscript entitled "Adherence to a healthy beverage score is associated with lower mortality in the adult population of Spain" for consideration by PLOS Medicine.

Your manuscript has now been evaluated by the PLOS Medicine editorial staff as well as by an academic editor with relevant expertise and I am writing to let you know that we would like to send your submission out for external peer review.

Please re-submit your manuscript within two working days, i.e. by Jul 18 2023 11:59PM.

Kind regards,

Philippa Dodd, MBBS MRCP PhD

PLOS Medicine

---

## [Decision Letter · Decision Letter 1]

11 Oct 2023

Dear Dr. Guallar-Castillón,

Thank you very much for submitting your manuscript "Adherence to a healthy beverage score is associated with lower mortality in the adult population of Spain" (PMEDICINE-D-23-01967R1) for consideration at PLOS Medicine. 

[LINK]

In light of these reviews, I am afraid that we will not be able to accept the manuscript for publication in the journal in its current form, but we would like to consider a revised version that addresses the reviewers' and editors' comments. Obviously we cannot make any decision about publication until we have seen the revised manuscript and your response, and we plan to seek re-review by one or more of the reviewers. 

We expect to receive your revised manuscript by Nov 01 2023 11:59PM. Please email us (plosmedicine@plos.org) if you have any questions or concerns.

We look forward to receiving your revised manuscript. 

Sincerely,

Philippa Dodd, MBBS MRCP PhD

PLOS Medicine

plosmedicine.org

COMMENTS FROM THE ACADEMIC EDITOR

I am unsure of the value because of some problematic scoring. This is not the first such study but one with a strong outcome. I do see some issues but let’s see how the authors respond.

COMMENTS FROM THE EDITORS

GENERAL

Please respond to all editor and reviewer requests detailed below in full.

The editorial team are in agreement that your manuscript could offer some contribution to the field but, we have concerns regarding novelty/advance and require that you clearly define this in the relevant parts of the manuscript.

We agree with the reviewers and the academic editor that clear justification for the scoring system you apply here as well as the advantages/disadvantages of the HBS as opposed to the HBI should be clearly detailed and discussed.

Your study is observational and therefore causality cannot be inferred. Throughout, please remove language that implies causality and refer to associations instead.

Please add line numbers starting at 1 on the title page and in continuous sequence throughout thereafter.

Please ensure that the study is reported according to the STROBE guideline, and include the completed STROBE checklist as Supporting Information. Please add the following statement, or similar, to the Methods: "This study is reported as per the Strengthening the Reporting of Observational Studies in Epidemiology (STROBE) guideline (S1 Checklist)."

When completing the checklist, please use section and paragraph numbers, rather than page numbers as these often change in the event of publication.

DATA AVAILABILITY

PLOS Medicine requires that the de-identified data underlying the specific results in a published article be made available, without restrictions on access, in a public repository or as Supporting Information at the time of article publication, provided it is legal and ethical to do so. Please see the policy at 

http://journals.plos.org/plosmedicine/s/data-availability and FAQs at 

http://journals.plos.org/plosmedicine/s/data-availability#loc-faqs-for-data-policy

PLOS defines the “minimal data set” to consist of the data set used to reach the conclusions drawn in the manuscript with related metadata and methods, and any additional data required to replicate the reported study findings in their entirety. Authors do not need to submit their entire data set, or the raw data collected during an investigation. Please submit the following data:

The values behind the means, standard deviations and other measures reported;

The values used to build graphs;

The points extracted from images for analysis.

The Data Availability Statement (DAS) requires revision. For each data source used in your study: 

COMPETING INTERESTS

All authors must declare their relevant competing interests per the PLOS policy, which can be seen here:

https://journals.plos.org/plosmedicine/s/competing-interests

For authors with ties to industry, please indicate whether any of the interests has a financial stake in the results of the current study.

TITLE

Please revise your title according to PLOS Medicine's style. Your title must be nondeclarative and not a question. It should begin with main concept if possible. "Effect of" should be used only if causality can be inferred, i.e., for an RCT. Please place the study design ("A randomized controlled trial," "A retrospective study," "A modelling study," etc.) in the subtitle (ie, after a colon).

ABSTRACT

Background:

Line 1 – suggest ‘relationship’.

Abstract Methods and Findings:

Please ensure that all numbers presented in the abstract are present and identical to numbers presented in the main manuscript text.

Please include the study design (observational cohort study), population and setting and, please clearly define the main outcome measures.

Please quantify the main results with 95% CIs and p values. Please report p as <0.001 and where higher the exact p values as p=0.002, for example. Please separate upper and lower bounds of 95% CIs with commas as opposed to hyphens as these can be confused with negative values.

Please include any important dependent variables that are adjusted for in the analyses.

Please include the actual amounts and/or absolute risk(s) of relevant outcomes not just relative risks or correlation coefficients. (example for absolute risks: PMID: 28399126). 

In the last sentence of the Abstract Methods and Findings section, please describe the main limitation(s) of the study's methodology.

Abstract Conclusions:

Please address the study implications without overreaching what can be concluded from the data; the phrase "In this study, we observed ..." may be useful.

Please interpret the study based on the results presented in the abstract, emphasizing what is new without overstating your conclusions.

Please avoid vague statements such as "these results have major implications for policy/clinical care". Mention only specific implications substantiated by the results.

In revising your conclusions, please ensure to avoid assertions of primacy ("We report for the first time....")

AUTHOR SUMMARY

At this stage, we ask that you include a short, non-technical Author Summary of your research to make findings accessible to a wide audience that includes both scientists and non-scientists. The authors summary should consist of 2-3 succinct bullet points under each of the following headings:

• Why Was This Study Done? Authors should reflect on what was known about the topic before the research was published and why the research was needed.

• What Did the Researchers Do and Find? Authors should briefly describe the study design that was used and the study’s major findings. Do include the headline numbers from the study, such as the sample size and key findings. 

• What Do These Findings Mean? Authors should reflect on the new knowledge generated by the research and the implications for practice, research, policy, or public health. Authors should also consider how the interpretation of the study’s findings may be affected by the study limitations. In the final bullet point of ‘What Do These Findings Mean?’, please describe the main limitations of the study in non-technical language.

Author Summary should immediately follow the Abstract in your revised manuscript. This text is subject to editorial change and should be distinct from the scientific abstract. Please see our author guidelines for more information: https://journals.plos.org/plosmedicine/s/revising-your-manuscript#loc-author-summary

INTRODUCTION

As above, please indicate whether your study is novel and how you determined that. If there has been a systematic review of the evidence related to your study (or you have conducted one), please refer to and reference that review and indicate whether it supports the need for your study.

Please conclude the Introduction with a clear description of the study question or hypothesis as in the current version.

METHODS and RESULTS

As above, please add the following statement, or similar, to the Methods: "This study is reported as per the Strengthening the Reporting of Observational Studies in Epidemiology (STROBE) guideline (S1 Checklist)."

Did your study have a prospective protocol or analysis plan? Please state this (either way) early in the Methods section.

For all observational studies, in the manuscript text, please indicate: 

(1) the specific hypotheses you intended to test, 

(2) the analytical methods by which you planned to test them, 

(3) the analyses you actually performed, and 

(4) when reported analyses differ from those that were planned, transparent explanations for differences that affect the reliability of the study's results. If a reported analysis was performed based on an interesting but unanticipated pattern in the data, please be clear that the analysis was data-driven.

As per reviewer comments (see below), please include further details of how this score differs from the HBI and the added advantage and please justify/explain your chosen scoring system/criteria.

Please define the length of follow up (eg, in mean, SD, and range).

As for the abstract, please quantify the main results with 95% CIs and p values. Please report p as <0.001 and where higher the exact p values as p=0.002, for example. Please separate upper and lower bounds of 95% CIs with commas as opposed to hyphens as these can be confused with negative values. 

Page 13 – ‘(95% CI, 0.55-0.88, P for linear trend=.003)’ please report as follows, in line with previous formatting requests/requirements, ‘(95% CI [0.55,0.88], p for linear trend=0.003),

TABLES

Thank you for indicating that your analyses are adjusted and detailing the factors adjusted for. To help facilitate transparent data reporting, please also include unadjusted analyses for comparison.

Please ensure that all abbreviations re defined in the caption/footnote including those used for statistical reporting – CI, HR, BMI, for example. Please check and amend throughout including in the supporting files where relevant.

Throughout, please separate upper and lower bounds of 95% CIs with commas as opposed to hyphens as the latter can be confused with the reporting of negative values.

Table 1 – the footnote appears to detail how the scoring system is applied to different beverages. This would be better presented as a table. Please revise accordingly and in line with the guidance above regarding the requirement for more nuanced details of the scoring system.

Table 2 (and others) – please report p as <0.001 as opposed to <.001. Please check and amend throughout all sections of the manuscript including text, tables, figures including in the supporting files.

FIGURES

Please ensure that all abbreviations including those used for statistical reporting are clearly defined in an appropriate caption/footnote.

Please ensure that p values are reported as <0.001 and where higher the exact p values as p=0.002, for example. 

Please replace uppercase ‘P’ with lowercase ‘p’.

Where adjusted analyses are presented to help facilitate transparent data reporting, please also present unadjusted analyses for comparison.

Where 95% CIs are reported, please also report p values as detailed above.

DISCUSSION

Please present and organize the Discussion as follows: a short, clear summary of the article's findings; what the study adds to existing research and where and why the results may differ from previous research; strengths and limitations of the study; implications and next steps for research, clinical practice, and/or public policy; one-paragraph conclusion. Please refrain from using sub-headings such that the discussion reads as continuous prose.

Please see reviewer #2 comments detailed below regarding wider discussion of the HBS which we agree with. 

Page 19 – please remove the sub-heading ‘conclusions’.

DECLARATIONS

Page 22 – please remove all declarations from the main manuscript and include only in the masncuript submission form. In the event of publication, these details will be compiled as metadata.

REFERENCES

PLOS uses the reference style outlined by the International Committee of Medical Journal Editors (ICMJE), also referred to as the “Vancouver” style. 

Please see our website for further guidelines here https://journals.plos.org/plosmedicine/s/submission-guidelines#loc-references

Journal name abbreviations should be those found in the National Center for Biotechnology Information (NCBI) databases. 

Please ensure that all web references include an ‘Accessed [date]’ as opposed to cited. Journal publications accessed on-line should not be cited as web references but as journal publications using the format detailed above.

SUPPORTING INFORMATION

As above, please include the completed STROBE checklist as Supporting Information. When completing the checklist please refer to section and paragraph numbers as opposed to page or line numbers as these often change in the event of publication.

S1 table – as for the tables in the main manuscript, please separate upper and lower CI bounds with commas as opposed to hyphens. Please replace ‘P’ with ‘p’ in the column header. Please report p <0.001 and where higher the exact p value as p=0.002, for example. Please ensure to amend the footnote also. To help facilitate transparent data reporting, please also present unadjusted analyses for comparison.

Comments from the reviewers:

Reviewer #1: Alex McConnachie, Statistical Review

The paper by Rodríguez-Ayala looks at the association between a derived healthy drinking score and mortality, using a large cohort study from Spain. This review considers the use of statistics in the paper.

Overall, the analysis is very good. The use of Cox PH models is appropriate, and it is nice to not have to remind the authors to check the PH assumption. Models with different levels of adjustment for confounders are reported. the exposure of interest is analysed both in categories but also as a continuous measure using cubic splines. A landmark analysis is done to check for reverse causality. The individual components of the main exposure variable are also assessed. Everything is explained well and presented clearly. My comments are fairly minor.

As ever, care should be taken to avoid language that implies a causal association, such as the use of the words "resulted in", as seen in the conclusion of the abstract.

Missing data are imputed using stochastic regression. What is "stochastic regression"? The reference is a book that I could not access. However, looking online, it appears to be another term for multiple imputation - if so, then this would be a more widely used term.

One of the results in Table 4 looked odd. For younger ages, the trend is nearly significant at p<0.05, with an increasing trend; for older ages, the trend is clearly decreasing. For Q3 and Q4, the confidence intervals are actually non-overlapping. Yet the interaction p-value is 0.65, which is a surprise. Could there be an error somewhere? The interaction p-values come from Wald tests, but I would normally use likelihood ratio tests in this situation. Do these give the same results?

There was a strange sentence in the discussion. "First, the number of deaths was low to evaluate cause specific mortality in subgroups of participants and sensitivity analyses were only performed with total mortality." As far as I can see, no analyses were done on cause specific mortality - everything was done with total mortality.

In Figure 1, what is the right-hand axis showing?

Reviewer #2: The authors explore the association of a Healthy Beverage Score with mortality risk in the ENRICA Study. Beverage consumption is an important domain of dietary intake that is often left out (with the exception of alcohol intake) of dietary guidance. Overall, the manuscript is well-written, and the study is of potential public health interest; however, there are some major concerns that should be addressed.

* As indicated in the text, the ENRICA study was drawn from a stratified cluster sample. Are the estimates provided weighted to be nationally representative? If so, please describe the sample weights and how they were applied. If not, was the enrolled sample in fact representative? Additional information on response rate is also needed.

* The text suggests that the HBS varies slightly from Duffey and Davy's Healthy Beverage Index HBI; however, the HBI is a 10-item index that includes water, total beverage energy, and fluid requirements in addition to the 7 items included in the current HBS. Please comment on these differences and how they likely impacted the scores and results.

* What is the rationale behind 4-point scoring system for components? It is unclear why alcohol, fruit juice, and artificially sweetened beverages are only assigned values 1 and 4. Scoring these components as either 1 or 4 appears to have resulted in some scores being more probable than others such that Q4 has almost 900 more individuals than Q3. Additionally, according to the footnote for table 2, the quartiles are very similar for men and women. If this is correct, what is reason for sex-specific quartiles?

* The alcohol consumption cutoff for women does not include women who drink 20-23 g/day. As it stands, it is <20 g/day for a score of 4 and >=24 g/day for a score of 1. Please redefine/clarify the alcohol consumption cutoff for women. Also, alcohol consumption (table 2) seems quite low in comparison to other beverage consumption. How is alcohol intake defined, as ethanol or volume of alcoholic beverages? were there a large proportion of zeros (i.e., nondrinkers) for certain beverages? If so, it would be more informative to show the median and IQR rather than mean (SD).

* It is unclear what the footnote under table 1 is specifying. Are these the scores of the participants within those tertiles and quartiles? 

* The cutoff values for the interaction tests appear to be designed to preserve sample size in strata but do not necessarily make sense if a biological interaction is suspected. The authors should consider more clinically meaningful categories or, if statistical power is the main concern, run models with continuous variables and their interactions. 

* Given the wide age range of the population, age-standardized descriptive statistics for table 2 would be more helpful in understanding what other covariates are potential confounders. For this reason (and given the strong potential of age to confound the HBS-mortality association), age should also be considered as the underlying time metric.

* Presently, one would expect multivariable adjustment for these variables to alter HR estimates; however, in Table 3, adjustment does not meaningfully shift estimates. This is surprising, given that coffee consumption, which tends to be highly correlated with smoking, is highlighted as explaining most of the association. 

* Table 4 - the number of deaths for each quartile do not equal the number of deaths provided in table 3. If correct, please explain why the cohort is different between the full analysis and stratified analyses. 

* Table S1: Why are there more deaths now in Quartile 1 and Quartile 3 than in the original analysis? Were the quartiles redefined for the sensitivity analysis? 

* There is no discussion on the actual HBS scale itself, only on its component beverages. The novelty of this study is the use of a HBS score, and advantages, implications, and limitations of the score should be more directly discussed.

Minor comment:

* Visually inspecting the Schoenfeld residuals is not a statistical test for the proportional hazards assumption. Please replace the word "test" with "assess". 

Reviewer #3: 

The article proposes the evaluation of the association of the adherence to a Healthy Beverage Score (HBS) and all-cause mortality in the ENRICA-1 cohort, a representative sample of the adult Spanish population. The article concerns an argument of current interest and is in general clear and well-written; the data come from a well-designed and well-conducted study with updated references. Few points should be addressed before considering for publication. 

1. In Figure 2, each item should be adjusted by the rest of the items that are part of the index (each beverage by the rest of the beverages).

2. Sensitivity analyses could be done for chronically ill patients to test if HBS also predicts mortality among the sickest.

3. Another sensitivity analysis would be to evaluate if the results are modified among those who have a low consumption of vegetables.

[LINK]

---

## [Decision Letter · Decision Letter 2]

14 Dec 2023

Dear Dr. Guallar-Castillón,

Thank you very much for re-submitting your manuscript "Association of a healthy beverage score with total mortality in the adult population of Spain: A nationwide prospective cohort study" (PMEDICINE-D-23-01967R2) for review by PLOS Medicine.

I have discussed the paper with my colleagues and the academic editor and it was also seen again by 2 reviewers. I am pleased to say that provided the remaining editorial and production issues are dealt with we are planning to accept the paper for publication in the journal.

[LINK]

We look forward to receiving the revised manuscript by Dec 21 2023 11:59PM.   

Sincerely,

Philippa Dodd, MBBS MRCP PhD

PLOS Medicine

plosmedicine.org

pdodd@plos.org

COMMENTS FROM THE ACADEMIC EDITOR

I think the authors have taken their data as far as they can. Their measure is really quite crude but pending the reviews, I am reluctantly ok with this. The authors really must be cautious in their findings. This points toward an impact but more refined measures are needed to totally test this relationship.

COMMENTS FROM THE EDITORS

GENERAL

Thank you for detailed responses to previous editor and reviewer comments. 

The Editorial team agree with the Academic Editor that you need to be cautious about how you report your findings. Specifically, we require greater consideration and contextualization of actual versus relative risk when reporting and discussing your results. Please see below for further comments which we require you address in full prior to publication.

TITLE

Please revise your title, specifically description as ‘prospective’ is misleading. Suggest the following, “Healthy beverage consumption and total mortality in Spanish adults: A nationwide cohort study”.

DATA AVAILABILITY STATEMENT

Thank you for updating your statement in line with point ‘b)’ below. Your statement requires revision. Please also indicate where these data are held i.e., where does the contact for data requests reside?

‘b) If the data are owned by a third party but freely available upon request, please note this and state the owner of the data set and contact information for data requests (web or email address). Note that a study author cannot be the contact person for the data.’

ABSTRACT

Line 29 – suggest removing ‘…the ENRICA-1 cohort…’

Line 31 – suggest making reference to the ENRICA-1 cohort here instead. Perhaps, ‘We conducted an observational cohort study leveraging data from the ENRICA-1 cohort, a study of community dwelling adults in Spain set up to…[please provide brief details]…’ . 

Please also define the acronym, ‘ENRICA-1’ at first use.

Line 38 – suggest moving sentence beginning ‘The HBS ranges from…’ to line 35 following ‘…(highest adherence)’.

Line 45 – please place CIs in brackets as follows, ‘[0.57, 0.91]’.

Line 48 – this statement is true of all observational studies. Please detail the limitations of the methodology as specific to your study.

Line 51 – ‘…associated with a one-quarter reduction in total mortality.’ This is an overstatement of your findings. Please revise and temper this statement (and later in your discussion, please see below). 

As above, it is vital that risk is contextualized appropriately in relation to actual and relative measures. While the reported percentage change is correct in proportional terms it does not reflect the true change in risk. Please amend (and throughout) this is a prerequisite to publication.

AUTHOR SUMMARY

Thank you for including an author summary. As written, it is lacking sufficient detail. Please revise.

Line 74 – this would be better placed under the section at line 79 when detailing what you did.

Please also revise this statement to improve clarity. Would it not be better to refer to the healthy beverage score (for consistency with the rest of the manuscript)? Suggest, ‘We describe a healthy beverage dietary score based on patterns of consumption of low-fat milk, coffee and tea, whole-fat milk, fruit juice, artificially-sweetened beverages, sugar-sweetened beverages and alcohol.’ Or similar

Line 79 – what did the researchers do and find? This section is very vague. It is impossible to understand how you conducted your study in whom or in how many when reading this section. Please revise including [in brief] the critical details.

INTRODUCTION

Final paragraph – it might be helpful to indicate that you develop a modified HBS (per methods section, line 158 onwards) as part of your study.

METHODS and RESULTS

Please see reviewer comments below regarding the description (and discussion later) of the HBI and HBS which we agree with. Please revise accordingly.

Throughout, please ensure that you do not overstate your findings. Please ensure adequate contextualization of the actual Vs relative risk. As reported the true risk could be over interpreted.

Line 140 - Ethics statement – can you please detail that this statement regarding consent applies to entry into the ENRICA-2 cohort study, as we understand things. Apologies if I have misunderstood but perhaps all the more reason to clarify!

TABLES

Throughout please replace ‘Men’ and ‘Women’ with ‘Male’ and ‘Female’ respectively. Please also ensure to amend footnotes/caption where relevant.

Please also see statistical reviewer comments below regarding table 2, which we agree with.

FIGURES

Figure 2 (forest plot) – please revise the presentation of p values which should be reported as <0.001 and where higher the exact p value as 0.442, for example. Please revise.

DISCUSSION

Please include a more nuanced discussion of the HBI Vs HBS as per reviewer comments below.

Line 376 – ‘In this large prospective population-based study…’ as for the title, please remove the word ‘prospective’, whilst the data from the ENRICA-1 cohort were collected prospectively, the data were analyzed retrospectively and as such your study cannot be described as a prospective study.

Line 378 – ‘the risk of death was reduced by a quarter’. As above please temper this statement and contextualize your findings in terms of actual risk – not just proportional change.

Line 380 – it be helpful to further quantify this statement based on the data generated.

Line 499 – please remove the word ‘remarkable’.

Line 501 – the phrase ‘on the other hand’ is in appropriately placed here. Please remove and perhaps replace with ‘in addition’, or similar.

Line 502 – what are the ‘vital statistics records’ you refer to here? This phrase has not been used elsewhere in the manuscript. Please revie and be in explicit.

Line 506 – as above the statement, ‘one quarter reduction in total mortality’ is an over interpretation and could be misleading, please revise.

SUPPORTING INFORMATION

Thank you for indicating that you analyses were preplanned. Did you document your analysis plan as part of funding proposal, for example? If so, please include a copy of the statistical analysis plan as supporting information.

Figures: S3 Figure (forest plot) – as for the main manuscript, please revise the presentation of p values which should be reported as <0.001 and where higher the exact p value as 0.442, for example.

Tables: Throughout please replace ‘Men’ and ‘Women’ with ‘Male’ and ‘Female’ respectively. For example, S2 table. Please also ensure to amend footnotes/caption where relevant.

SOCIAL MEDIA

To help us extend the reach of your research, please detail any X (formerly Twitter) handles you wish to be included when we tweet this paper (including your own, your coauthors’, your institution, funder, or lab) in the manuscript submission form when you re-submit the manuscript.

COMMENTS FROM THE REVIEWERS:

Reviewer #1: Alex McConnachie, Statistical Review

I thank the authors for their consideration of my previous comments. I am satisfied with their responses. My remaining observations are very minor.

I note that Table 2 has been changed. It now shows standard errors for the mean of the continuous measures. I am not keen on this, since the aim of this table should be to show the distribution of the data, whereas the SE is a measure of precision. Could these SE's be replaced with standard deviations, or simply omitted, showing the mean only?

Incidentally, the values reported as the SE for mean age do not look right - they seem far too large.

There is an oddly phrased sentence on lines 308-309: "…with lower mortality among those who were chronically ill" - I believe something like "with higher HBS associated with lower mortality among those who were chronically ill, but no association in those without a chronic illness" would be more accurate.

Reviewer #2: The authors were generally responsive to reviewer comments. I have a few additional questions and comments for their consideration.

1. I disagree with the authors that the HBS varies "slightly" from HBI as there are a handful of differences that could each be considered substantial. I think the methods could simply cite the HBS applied by Hu et al and the HBI could be left to the discussion.

2. The authors' response to my question about the scoring rationale suggests that had there been a wider distribution of intake for either fruit juice or artificially sweetened beverages then they would have been scored similarly to the other items. It should be noted in the methods that lack of consumption of these beverages is the reason for the difference in scoring.

3. No where in the methods or tables do the authors define alcohol as grams of ethanol (rather than grams of alcoholic beverages). Given that all other beverages are defined based on the total beverage, and not just a component, I would expect the authors to add this detail.

4. The authors now use age the underlying time metric but include age as a confounder in the model. Though it wouldn't impact HRs much (if at all), I am unsure if this is a typo in the tables or in fact how the model was constructed. Also, it should be added to the footnotes that age is the underlying time metric as model 1 does account for age if this is the case.

5. The authors observe an inverse association among those with a chronic disease but not among those without. In line with this finding, there is an inverse association in older but not younger individuals. Given, the wide age range in this study, did the authors look at how the leading causes of deaths differed in these subgroups? Could it be that the HBS is important for chronic disease-specific death only? This is alluded to in the discussion as the potential mechanism discussed are related to chronic disease. The subgroup analyses are also somewhat at odds with the interpretation of HBS being inversely associated with "premature" mortality or mortality in the "general population".

6. The abstract and discussion state "a one-quarter reduction in total mortality" this is misleading as results in this analysis are relative and not absolute risk reductions.

[LINK]

---

## [Editor Report · Decision Letter 3]

21 Dec 2023

Dear Dr Guallar-Castillón, 

On behalf of my colleagues and the Academic Editor, Professor Barry Popkin, I am pleased to inform you that we have agreed to publish your manuscript "Association of a healthy beverage score with total mortality in the adult population of Spain: A nationwide cohort study" (PMEDICINE-D-23-01967R3) in PLOS Medicine.

PRESS

Thank you again for submitting to PLOS Medicine, it has been a pleasure handling your manuscript. We look forward to publishing your paper. 

Best wishes,

Pippa 

Philippa Dodd, MBBS MRCP PhD 

PLOS Medicine

pdodd@plos.org